# Microfluidic Single-Cell Manipulation and Analysis: Methods and Applications

**DOI:** 10.3390/mi10020104

**Published:** 2019-02-01

**Authors:** Tao Luo, Lei Fan, Rong Zhu, Dong Sun

**Affiliations:** 1Department of Biomedical Engineering, City University of Hong Kong, Hong Kong, China; taoluo4-c@my.cityu.edu.hk (T.L.); leifan-c@my.cityu.edu.hk (L.F.); 2State Key Laboratory of Precision Measurement Technology and Instruments, Department of Precision Instrument, Tsinghua University, Beijing 100084, China; zr_gloria@mail.tsinghua.edu.cn; 3Shenzhen Research Institute of City University of Hong Kong, Shenzhen 518057, China

**Keywords:** microfluidics, single-cell manipulation, single-cell analysis

## Abstract

In a forest of a hundred thousand trees, no two leaves are alike. Similarly, no two cells in a genetically identical group are the same. This heterogeneity at the single-cell level has been recognized to be vital for the correct interpretation of diagnostic and therapeutic results of diseases, but has been masked for a long time by studying average responses from a population. To comprehensively understand cell heterogeneity, diverse manipulation and comprehensive analysis of cells at the single-cell level are demanded. However, using traditional biological tools, such as petri-dishes and well-plates, is technically challengeable for manipulating and analyzing single-cells with small size and low concentration of target biomolecules. With the development of microfluidics, which is a technology of manipulating and controlling fluids in the range of micro- to pico-liters in networks of channels with dimensions from tens to hundreds of microns, single-cell study has been blooming for almost two decades. Comparing to conventional petri-dish or well-plate experiments, microfluidic single-cell analysis offers advantages of higher throughput, smaller sample volume, automatic sample processing, and lower contamination risk, etc., which made microfluidics an ideal technology for conducting statically meaningful single-cell research. In this review, we will summarize the advances of microfluidics for single-cell manipulation and analysis from the aspects of methods and applications. First, various methods, such as hydrodynamic and electrical approaches, for microfluidic single-cell manipulation will be summarized. Second, single-cell analysis ranging from cellular to genetic level by using microfluidic technology is summarized. Last, we will also discuss the advantages and disadvantages of various microfluidic methods for single-cell manipulation, and then outlook the trend of microfluidic single-cell analysis.

## 1. Introduction

Over the past few decades, cellular heterogeneity has gradually been emphasized on fundamental biological and clinical research as numerous novel tools/methods for single-cell analysis have emerged [1]. Phenotype heterogeneity between genetically identical cells plays an important role in tumor metastasis [2], drug resistance [3], and stem cell differentiation [4]. For instance, different responses of individual cells to drugs cause the emergence of drug-resistant cells, but only a small percentage (0.3%) of these cells have the ability for tumor recurrence [5]. However, cellular heterogeneity has been masked for a long time because previous biological studies are mainly based on manipulating and analyzing cells at the bulk-scale, which interpreted all phenomena by using average results. Until today, single-cell study still has been recognized as the most straightforward way to performance comprehensive heterogeneity study from the aspects of cellular behavior to genetic expression. Comprehensive single-cell study heavily relies on the use of high-throughput and efficient tools for manipulating and analyzing cells at the single-cell level.

Single-cell analysis is technically more difficult than bulk-cell analysis in terms of the sizes of cells and the concentrations of cellular components. The majority of cells, such as mammalian and bacteria cells, have sizes at the scale of microns. Therefore, manipulation of those cells at the single-cell level becomes difficult when using traditional biological tools, such as petri-dishes and well-plates. Additionally, most of the intracellular, extracellular components are presented in very small concentrations and have a wide range of concentrations, which demand highly sensitive and specific detection methods. Many single-cell analysis applications require a single-cell isolation first, and multi-well plates are commonly used in most biological labs for single-cell isolation, which is low in efficiency and labor-intensive [6]. While the use of robotic liquid handling workstation reduces the labor intensity, it is very expensive for some labs to afford it [7]. Flow cytometry or laser scanning cytometry, which rapidly screens fluorescently labeled cells in a flow, has been developed and recognized as a golden standard for single-cell analysis for a long time [8]. Taking flow cytometry as an example, although they are automatic, capable of multiple detections, and efficient in single-cell sorting, they are bulky, mechanically complicated, expensive, and demanding for relatively large sample volumes. Besides, they can only be used for analyzing cells at one time-point. Hence, it is impossible to use flow cytometry for continuously monitoring cell dynamics. Owing to the capability of manipulating and controlling fluids in the range of micro to pico-liters, microfluidics has been developed as a platform-level and continuously evolving technology for single-cell manipulation and analysis for about two decades.

Microfluidics has many incomparable advantages over conventional techniques. Firstly, the microfluidic chip can be flexibly designed to fulfill the demands of diverse single-cell manipulation and analysis tasks. For instance, single-cell manipulation can be achieved by using either passive [9,10,11] or active [12,13] method, and single-cell analysis can be achieved by implementing either optical [14,15] or electrochemical [16,17] method. Secondly, miniaturized microfluidic systems work can work with very small volume (down to pL level) of liquid, which helps to reduce sample loss and decrease dilution, resulting in highly sensitive detections. Hence, numerous microfluidics-based biosensors have been developed. Thirdly, microfluidics allows for high-throughput parallel manipulation and analysis of the sample, which is beneficial for the statistically meaningful single-cell analysis. Fourthly, multiple functionalities are easily integrated on the same chip, which allows for automation, and can also avoid contamination and errors introduced by manual operations. Many single-cell studies require single-cell capture/isolation, and different microfluidic methods, such as hydrodynamic [11,18,19], electrical [20], optical [21], magnetic [22], and acoustic [23] methods, have been developed. Various detection methods, such as fluorescence microscopy, fluorometry and mass spectroscopy, can be combined with microfluidic systems for single-cell analysis from cell morphology to secreted proteins. As for either single-cell manipulation or single-cell analysis, it is hard to obtain a comprehensive result by merely using one method. Therefore, two or more methods are usually combined into a microfluidics system for various single-cell studies [24,25].

While reviews about single-cell manipulation and analysis by using microfluidics are reported almost every year, systematic summarization of this area can give valuable references to both academic and industrial fields. In this review, we mainly focus on microfluidic technologies for single-cell manipulation analysis from the aspects of methods and applications. We highlight methods that are promising for future development, which are discussed in terms of single-cell manipulation including hydrodynamic, electrical, optical, magnetic, acoustic, and micro-robots assisted methods. We also highlight applications that are accepted by academic and industrial fields, which are discussed in terms of single-cell analysis from cellular to protein analysis. Last, we also discuss the technology and application trend for microfluidics based single-cell analysis.

## 2. Microfluidic Single-Cell Manipulation

With the development of Microelectromechanical Systems (MEMS) technology, many micro-scale devices have been fabricated for bioanalysis at single-cell resolution. As a powerful technology to perform precise fluidic control, microfluidics has attracted great interests for various single-cell manipulations, such as single-cell encapsulation and single-cell trapping (Table 1). Nowadays, various methods, including hydrodynamic, electrical, optical, acoustic, magnetic, and micro-robotic method, were used for diverse microfluidic single-cell manipulations.

### 2.1. Hydrodynamic Method

Compared with other single-cell manipulation methods, the hydrodynamic method is much simpler and higher throughput. Hydrodynamic manipulation mainly relies on the interaction among microstructure, fluid, and cells. This technique has the advantages of high throughput, less damage to cells, mature chip fabrication crafts, and easy integration with other analysis functionalities. Based on the used mechanisms, the hydrodynamic method can be categorized into droplet microfluidics, inertial microfluidics, vortex, and mechanical methods.

#### 2.1.1. Droplet Microfluidics

Droplet microfluidics has attracted more and more interests for its capability to encapsulate cells and many reagents in a microscale environment [47]. It is also a powerful technique for high-throughput single-cell encapsulation. The size, shape, and uniformity of droplets can be precisely controlled. This method usually requires two immiscible fluids to create monodispersed water-in-oil (w/o) microdroplets with sizes that range from the submicron to several hundreds of microns [48]. As shown in Figure 1, three types of microfluidic droplet generation approach usually used: T-junction, flow focusing, and co-flow [28]. However, the basic principles of these three approaches are the same. One fluid becomes the dispersed phase to form the droplets and the other fluid becomes the continuous phase to separate the droplets. As illustrated in Figure 1a, Zhang et al. integrated T-junction structure with droplet inspection, single-cell droplet sorting and exporting on one chip to analyze DNA and RNA at both gene-specific and whole-genome levels [49]. Zilionis et al. utilized flow focusing structure, shown in Figure 1b, to conduct single-cell barcoding and sequencing [27]. Adams et al. encapsulated multi-component in one droplet using co-flow method, as shown in Figure 1c [50]. All three approaches require to control each fluid phase precisely, which usually make the system a little bit complex. Khoshmanesh et al. proposed a novel mechanism for generating microscale droplets of aqueous solutions in oil using a highly porous PDMS sponge [51]. Compared to the existing microfluidic droplet generation approach, the sponge-based approach is a self-sufficient, stand-alone device, which can be operated without using pumps, tubes, and microfluidic skills. Single-cell is randomly encapsulated in each droplet, and the number of cells in each droplet follows nonuniform poisson distribution. Therefore, the cell suspension usually requires to be highly diluted before encapsulation to ensure only one single cell in one droplet. However, this procedure leads to reagent waste and degrades throughput because most of the generated droplets do not contain single cells. Several methods have been developed to overcome this limitation. For example, post-sorting based on property differences can be applied to enhance single-cell encapsulation efficiency. Moreover, this technique is hampered by some other drawbacks. For instance, the cell culture in droplet is suspended, which means that the culture of adherent cell in droplet is difficult. Introducing/picking components in/out of droplets without risking cross-contamination among cell-contained droplets is difficult.

#### 2.1.2. Inertial Microfluidics

As a cross-streamline cell manipulation method, inertial microfluidics is based on using inertial forces under certain high flow rates to continuously focus and sort cells with different sizes and shapes, as illustrated in Figure 2a [52]. The inertial microfluidics has the advantages of simple chip structure, ultra-high throughput, and less damage to cells which is beneficial to maintain high cell survival rate for downstream culture. However, intercellular interaction can greatly reduce the efficiency of cell manipulation by using inertial microfluidics. Therefore, it is only applicable to work under a certain cell concentration. This method is usually used for separating single circulating cancer cells from blood cells, and it usually requires pre-dilution of blood samples [53,54]. Nathamgari et al. used inertial microfluidics to separate the single neural cells and clusters from a population of chemically dissociated neurospheres shown in Figure 2a [55]. In contrast to previous sorting technologies that require operating at high flow rates, they implemented a spiral microfluidic channel in a novel-focusing regime that occurs at relative lower flow rates. The curvature-induced Dean’s force focused the smaller single cells towards the inner wall and the larger clusters towards the center of channel.

#### 2.1.3. Vortex

Vortex based method is based on generating a time-averaged secondary flow known as steady streaming eddies [56,57], which is generated by the interaction between frequency oscillations of the fluid medium and fixed cylinder in a microchannel to trap single cells. Therefore, the vortex approach is also called single-cell hydrodynamic tweezer. In 2006, Lutz et al. used motile phytoplankton cells to measure the trapping location and trapping force [34]. They proved that each eddy traps a single-cell near the eddy center, precisely at the channel midplane, and the trapped cell is completely suspended by the fluid without touching any solid surface. Furthermore, the trapping force are comparable to dielectrophoretic force and optical tweezer force, whereas the trap environment is within physiological limits of shear in arterial blood flow. Thus, the hydrodynamic tweezer does not limit the cell type, shape, density, or composition of the fluid medium. As shown in Figure 2b, Hayakawa et al. adopted three micropillars arranged in a triangular configuration and an xyz piezoelectric actuator to apply the circular vibration to generated vortex for trapping and 3D rotating mouse oocytes single-cell [58]. Additionally, they measured the rotational speeds in the focal and vertical planes as 63.7 ± 4.0°∙s^−1^ and 3.5 ± 2.1°∙s^−1^, respectively.

#### 2.1.4. Mechanical Method

Mechanical method for single-cell manipulation mainly refers to the use of membrane pump, microvalve, and microstructures.

●  Microvalve

Having the advantages of small size, fast response, and simple fabrication, the active microvalve is widely used in the manipulation of single cells [29]. Multilayer soft lithography, which is the basement of microvalve, was first developed by Quake’s group [59]. They used this technique to build active microfluidic systems containing on-off valves and switching valves. The invention of these valves made it possible to realize high density fluid control and large-scale functional integration on a chip, which is a milestone in the development of microfluidic technology. Thus, this kind of microvalve is called “Quake valve”. The structure of “Quake valve” consists of a vertically crossed flow layer and a control layer, between which a deformable film is formed. When the pressure is applied through the control layer, the film deforms and sticks to the bottom surface of flow channel to block the flow in the flow layer. On the contrary, when the control layer does not exert pressure or apply a relatively small pressure, the flow layer appears to be open or semi-open. Fluidigm Company of United States developed a single-cell automatic pretreatment system called C1 Single-Cell Auto Prep System based on this technology, which can automatically separate 96 suspended cells at one time. At present, the system has been used in many universities and research institutions in the world for single-cell genomics research. Shalek et al. uses this system to isolate single-cell and conduct RNA-seq libraries prepared from over 1700 primary mouse bone-marrow-derived dendritic cells shown in Figure 3 [60]. While microvalves offer many advantages, their external control devices are extremely complex and cumbersome. Thus, the external operation of large-scale integrated microfluidic chips based on microvalves must be simplified and adapted to the working habits of biological researchers.

●  Microstructure

The method based on microstructures, such as microtraps and microwells, only requires researcher to design microstructures whose sizes are similar to the size of a single cell. After injecting cells into microfluidic chip, cells flow along the streamlines of the laminar flow, and can be trapped by shearing force generated by microstructure. Current microstructure for single-cell trap include U-shaped, S-shaped, and microwell based traps.

U-shaped microtrap was demonstrated firstly by Di Carlo et al. to trap and culture single Hela cells [61]. The U-shaped structure was fabricated by polydimethylsiloxane (PDMS), bonded with glass and there was a gap between the trap and the glass substrate for increasing the single-cell trap efficiency. Then, the U-shaped microtrap was modified by Wlodkowic et al. to add three gaps in the edge of U-shaped groove for increasing the cell viability [19]. In the last years, Tran et al. designed U-shaped micro sieve comprised semicircular arcs spaced at specific offsets and distance as shown in Figure 4a [62]. Using this micro sieve, they realized label-free and rapid human breast cancer single-cell isolation with up to 100% trapping yield and >95% sequential isolation efficiency. Luo et al. fabricated a high throughput single-cell trap and culture platform to investigate clonal growth of arrayed single-cells under chemical/electrical stimuli for the week-scale period [63]. To achieve deterministic single-cell capture in large-sized microchambers, a U-shaped sieve with a 5μm-thick bottom microchannel was used to capture a single-cell, and two stream focusing arms were placed in front of the sieve to enhance capture efficiency. Zhang et al. demonstrated a handheld single-cell pipette, which allows for rapid single-cell isolation from low concentration cell suspension, by using a U-shaped microtrap, as shown in Figure 4b [37].

S-shaped microstructure is based on the different fluidic resistance in different position of microchannel. The S-shaped microstructure was firstly reported by Tan et al. to trap and release microbeads [64]. After microbead trapping, one objective bead could be release by optical microbubbles. In recent years, S-shaped microstructure was also further optimized. Kim. et al. proposed a simple, efficient S-shaped microfluidic array chip integrated with a size-based cell bandpass filter [65]. The key advancement to this chip is not the optimization of single-cell trap, but the capability of trapping cells within a specific range of sizes. Mi et al. combined U-shape and S-shape, which is named m-by-n trap units, to pattern single Hela cells, as shown in Figure 4c [10]. Each unit has two roundabout channels and one capture channel. Different from previous S-shaped microchannels, this structure enables each trap unit to be treated equally and independently. Therefore, any unit can be selected for finalizing the geometric parameters of the fluidic channels to satisfy the capture condition.

Microwell is another passive microstructure for single-cell trapping. This structure is mainly based on the size match of the single cell and microwell. When the diameter of microwell approaches the diameter of a single cell, redundant cells will be wash out and one single cell settled down to one microwell can be trapped. After injecting cell suspension to microfluidic channel, cells will sediment down and go into microwells. When the depth of the microwell is deep enough, vortex generates inside the microwell and single cells can be trapped firmly. Most of current microwells were fabricated by soft lithography. In 2004, Revzin et al. developed a cytometry platform for characterization and sorting of individual leukocytes [32]. Poly (ethylene glycol) (PEG) was employed to fabricate arrays of microwells composed of PEG hydrogel walls and glass substrates. PEG micropatterned glass surfaces were further modified with cell-adhesive ligands, poly-L-lysine, anti-CD5, and antiCD19 antibodies. Cell occupancy reached 94.6 ± 2.3% for microwells decorated with T-cell specific anti-CD5 antibodies. Later on, Rettig et al. fabricated tens of thousands of microwells on a glass substrate [9]. And they characterized microwell occupancy for a range of dimensions and seeding concentration using different cells. For culturing single-cell in one chamber, Lin et al. fabricated a dual-well (DW) device which allows for highly efficient loading of single-cells into large microwells for single-cell culture, as shown in Figure 4d [66]. Single-cell loading in large microwells is achieved by utilizing small microwells to trap single cells followed by using gravity to transfer the trapped single cells into large microwells for single-cell culture.

### 2.2. Electrical Method

The electrical methods have been widely used to trap and pattern single-cell because it usually imposes lower physical pressure on the cell membrane. Two typical methods exist for electronically controlled single-cell manipulation: Dielectrophoresis (DEP) and electroosmosis.

#### 2.2.1. Dielectrophoresis (DEP)

DEP manipulation relies on the use of DEP forces, which are generated by the interaction between the nonuniform electric field and the cells. DEP forces applied to cells depend on the size of the cells, the dielectric properties of the cells and the surrounding solution, the gradient of the electric field, and the frequency of the electric field [67]. As shown in Figure 5a, DEP forces can be categorized as positive or negative [68]. Under positive DEP forces, cells move to strong electric field regions. By contrast, under negative DEP forces, cells move to weak electric field regions. The frequency of the electric field when the DEP force is zero is called cross-over frequency, at which the DEP forces applied to cells is zero. The cell manipulation performance of the DEP chip depends largely on the design of the DEP electrode. DEP can be easily combined with microfluidic systems, is label-free, and has high selectivity in manipulating rare cells. Most DEP cell manipulation systems require a low-conductivity solution. However, physiological solutions, such as blood and urine, are highly conductive. Thus, cell samples require stringent pretreatment. This requirement limits the application of DEP-based approaches and may have prevented the application of DEP-based cell manipulation in the clinical field. Taff et al. first presented a scalable addressable positive-dielectrophoretic single-cell trapping and sorting chip using MEMS technology based on silicon substrate [31]. The chip incorporates a unique “ring-dot” pDEP trap geometry organized in a row/column array format. A passive, scalable architecture for trapping, imaging, and sorting individual microparticles, including cells, using a positive dielectrophoretic (pDEP) trapping array was fabricated. Thomas et al. presented a novel micron-sized particle trap that uses nDEP to trap cells in high conductivity physiological media [13]. The design is scalable and suitable for trapping large numbers of single-cells. Each trap has one electrical connection, and the design can be extended to produce a large array. The trap consists of a metal ring electrode and a surrounding ground plane, which creates a closed electric field cage in the center. The device is operated by trapping the single latex spheres and HeLa cells against a moving fluid. In recent years, Wu. et al., as shown in Figure 5b, reported a design and fabrication of a planar chip for high-throughput cell trapping and pairing by pDEP within only several minutes [20]. The pDEP was generated by applying an alternating current signal on a novel two-pair interdigitated array (TPIDA) electrode. In Figure 5c, Huang et al. reported DEP-based single-cell trap and rotation chip for 3D cell imaging and multiple biophysical property measurements [69]. They firstly trapped a single-cell in constriction and subsequently released it to a rotation chamber formed by four sidewall electrodes and one transparent bottom electrode, which are powered by AC signals.

#### 2.2.2. Electroosmosis

Electroosmotic flow is caused by the Coulomb force that is induced by an electric field on net mobile electric charge in a solution. Two kinds of electroosmosis are usually used for cell manipulation, namely, alternating current electroosmosis (ACEO), and induced charge electroosmosis (ICEO). ACEO is induced by ionic cloud migration in response to a tangentially applied electric field on the electrode surface and only occurs when the applied frequency is far below the charge relaxation frequency of the fluid. ACEO is one of the most promising electrokinetic approaches for developing fully integrated lab-on-a-chip systems because it is a label-free and well-established technique for microelectrode fabrication, as well as low voltage requirement. Gilad Yossifon et al. reported a multifunctional microfluidic platform on-chip electroporation integrated with ACEO-assisted cell trapping [71]. ACEO vortices enable the rapid trapping/alignment of particles at sufficiently low activation frequencies. ICEO is an electrochemical effect that occurs on the surface of an object and manifests as nonlinear fluid flow under electric conditions. As shown in Figure 5d, the induced charge diffusion stimulated by the electric field induces slip-like local fluid flow under the applied electric field. The induced diffusion charges are distributed in the boundary layer on the solid/liquid interface, i.e., a double electrical layer. Electric force can be applied to the fluid molecules to control microfluidic movement given the existence of the double layer [72]. ICEO efficiently enriches cells in specific areas, and high-throughput and noncontact single-cell capture can also be achieved in a given area (Figure 5d) [70]. The ICEO microfluidic chip can enable the small-scale integration of electrode arrays and microchannels. The ICEO chip has a compact structure and is easy to process. Given that the floating electrode does not require an external electrical signal wire connection, the design layout can be flexibly designed in accordance with different application scenarios. However, the performance of the ICEO microfluidic chip is dependent on the induced secondary flow. In the case of continuous sampling, cell manipulation efficiency and precision will greatly decrease with the increase of the flow rate.

### 2.3. Optical Method

Three types of optical cell manipulation methods currently exist: Optical tweezer, optically induced-dielectrophoresis (ODEP), and opto-thermocapillary.

#### 2.3.1. Optical Tweezer

The use of optical tweezers to move particles was first discovered by the American scientist Arthur Ashkin. He found that a highly focused laser beam could drag an object with a higher refractive index than the medium to the middle of the laser beam. He first studied the “optical tweezer” effect with a single laser beam in 1987 [73]. The tweezer based on monochromatic lasers can manipulate particles within the size range of nanometers to tens of microns. Therefore, the optical tweezer can be used to manipulate biological single-cells. Robotic-assisted optical tweezers have enabled the automated multidimensional manipulation of cells and have been applied in studies on cell mechanics [74], cell transportation [42], and cell migration [75]. Optical tweezers can achieve cell manipulation under static environments and combined with microfluidic chips to realize cell manipulation under continuous flow. The most typical application is optical tweezer-enhanced microfluidic cell sorting [12]. As shown in Figure 6a, the cell sample is first focused on the channel upstream through the sheath flow. Then, the cells are identified through the image processing of the fluorescence characteristics of the cells. The tweezer automatically captures and drags the target single-cells to make them laterally cross the streamline. The target single-cells are thus collected at a specific outlet.

Optical tweezers offer the advantages of high accuracy, non-intrusiveness, and high-throughput single-cell manipulation. However, the manipulative laser force exerted on the cells is typically in the order of pico-Newtons. Thus, cell manipulation under continuous flow requires low fluid velocity; otherwise, the optical tweezers will experience difficulty in deflecting cells. In addition, the massive peripherical optical system required by this technique is difficult to miniaturize and is also very expensive.

#### 2.3.2. Optically Induced Dielectrophoresis (ODEP)

Optically induced dielectrophoresis (ODEP) is a novel particle manipulation technology. The forces used to manipulate particles in ODEP-based chip are the same as those used in traditional DEP technology. That is, a nonuniform electric field is used to polarize cells and generate DEP forces. The difference between the two technologies lies in the technique by which the nonuniform electric field is generated. In contrast to conventional DEP manipulation, ODEP does not need prefabricated electrode patterns. However, digital micromirror device (DMD) projector can be used to project the light pattern onto the chip substrate via a microscope to generate flexible and controllable virtual electrodes [21]. Its working principle is similar to that of photovoltaic power generation. An amorphous silicon substrate material generates photocarriers under light excitation, thereby increasing carrier concentration in the illumination area. Consequently, the electrical conductivity of the illumination area rapidly increases, thereby generating a nonuniform electric field. ODEP can also enable the DEP manipulation of live single-cell. It is contactless and label-free. As shown in Figure 6b, Xie et al. utilized ODEP to effectively trap and transport unicellular swimming algae [76]. They found that the trapped cells started to rotate and demonstrated that functional flagella played a decisive role in the rotation. Furthermore, they also realized homodromous rotation of a live C. reinhardtii cell array in an ODEP trap and the speed of rotation can be controlled by varying the optical intensity.

The ODEP system is considerably simpler than optical tweezer systems and can be miniaturized. Moreover, ODEP can manipulate cells that are not optically transparent, thus exhibiting great flexibility. However, ODEP and its clinical applications are hindered by the same inherent drawbacks as traditional DEP: The manipulation of cells in low-conductivity solutions. Moreover, the substrate of the ODEP chip is opaque because of the deposition of amorphous silicon. The opacity of the substrate precludes the use of an inverted biological microscope for live cell imaging.

#### 2.3.3. Opto-Thermocapillary

Different from the above two optical methods, the opto-thermal method uses light to generate heat for opto-thermophoresis or opto-thermocapillary. Thermophoresis is the thermos-migration or thermos-diffusion of particles subjected to a temperature gradient [78]. The opto-thermophoresis can be used to trap small biological molecules. However, it is difficult for trapping large particle, such as cells. The thermocapillary, also named thermal Marangoni effect, refers to mass transfer along a liquid–gas interface due to a surface tension gradient created by a temperature gradient. Thus, opto-thermocapillary actuation is not dependent on the optical properties of the object. And opto-thermocapillary is not sensitive to the electrical properties of the liquid medium and the object. However, opto-thermocapillary actuation shares the flexibility of optical control, which enables parallel and independent manipulation of multiple micro-objects [79]. Opto-thermocapillary force can be used to actuate microbubbles that enable manipulating single-cells and biomolecules [80]. As shown in Figure 6c, a microbubble can be generated after focusing optical beam on the absorbing coating. This microbubble can be seen as micro-scale actuator to manipulate single cell. Hu et al. used a near-infrared laser focused on indium tin oxide (ITO) glass to generate thermocapillary effect that can trap and transport living single cells with forces of up to 40 pN [77]. Moreover, they also patterned single-cell in two kinds of hydrogels: Polyethylene glycol diacrylate (PEGDA) and agarose. High viability rates were observed in both hydrogels, and single cells patterned in agarose spread and migrated during culture.

### 2.4. Acoustic Method

Acoustic cell manipulation is based on the complex flow–structure interaction that occurs when acoustic waves enter a microfluidic channel. Acoustic waves can be categorized into body and surface waves. A surface acoustic wave (SAW) is an elastic acoustic wave that can propagate only on the substrate surface. Most of its energy is concentrated on the substrate surface at a depth of several wavelengths. Given their advantages of high frequency, high energy density, good penetrability, and easy integration, SAW chips have been widely used in recent years for cell manipulation on microfluidic chips. Interdigital transducers (IDTs) are generally used to generate SAW. The resonant frequency of the SAW can be controlled by adjusting the interdigital spacing of electrodes. The resonant frequency of SAW devices can reach up to GHz, indicating that they can precisely control micron-sized or even submicron-sized particles. In addition, the distribution of the acoustic field can be regulated by changing the shape of the IDTs, further demonstrating the flexibility of this manipulation method. SAW devices are generally processed using standard microfabrication technology. Thus, SAW devices have excellent reproducibility and consistency. The planar processing method enables the integration of SAW devices into microfluidic chips. When acoustic waves propagate into a fluidic medium, the fluid acquires momentum by absorbing acoustic waves. Bulk flow, in turn, is induced by acoustic wave absorption. This phenomenon is called the acoustic flow effect. The particle moves with the fluid if its size is substantially smaller than the wavelength and its density is small. This phenomenon thus enables the manipulation of particles in the fluid. When two columns of SAWs propagate opposite each other on the same surface, a standing surface acoustic wave (SSAW) is generated. Under the influence of SSAW, particles are subjected to standing wave acoustic radiation force and then accumulate at the antinode or node position depending on the properties of the particles and the surrounding medium, such as density and compressibility. Acoustic radiation is mainly attributed to the effects of particles on sound waves. These effects include reflection, refraction, and absorption and result in the exchange of momentum between sound waves and particles. The magnitude of the acoustic radiation force is related to the physical properties, such as wavelength and amplitude, of the SSAW and the size and density of the particles. In recent years, considerable research effort has been directed toward the establishment of SSAWs on microfluidic chips to realize precise and high-throughput single-cell manipulation. As shown in Figure 7a, Collins et al. introduced multiple high-frequency SSAWs with one cell per acoustic well for the patterning of multiple spatially separated single-cells [23]. They also characterized and demonstrated patterning for a wide range of particle sizes, and patterning of cells, including human lymphocytes and red blood cells infected by the malarial parasite Plasmodium falciparum.

Acoustic technology for single-cell manipulation presents the advantages of high frequency, high energy density, good penetrability, easy fabrication, easy integration, and noninvasiveness. However, the application of acoustic control in microfluidic single-cell manipulation remains in its infancy and poses numerous problems that still require resolution. These problems include nonlinear interactions among acoustic waves, fluids, and cells. Moreover, current acoustic chips are mostly based on LiNbO_3_ substrates, and device substrates based on new piezoelectric materials must be investigated.

### 2.5. Magnetic Method

Magnetic manipulation refers to the manipulation of cells through using permanent magnets or electromagnets. This method requires the surface label of cells with immunomagnetic beads because cells typically lack paramagnetic or diamagnetic properties. Surface-modified nanomagnetic beads adhere onto the cell surface through specific interactions between the antibody and the antigen. Given this requirement, the magnetic method is a label-based approach. As shown in Figure 7b, Shields IV et al. developed a magnetic microfluidic platform comprised of three modules that offers high throughput separation of cancer cells from blood and on-chip organization of those cells for streamlined analyses [81]. The first module uses an acoustic standing wave to rapidly align cells in a contactless manner. The second module then separates magnetically labeled cells from unlabeled cells, offering purities exceeding 85% for cells and 90% for binary mixtures of synthetic particles. Finally, the third module contains a spatially periodic array of microwells with underlying micromagnets to capture individual cells for on-chip analyses.

Cell sample manipulation using immunomagnetic beads is simple and reliable, and its adoption in many large-scale analytical platforms for clinical applications has matured. However, the application prospects of this method are limited, given that it encounters difficulty in simultaneously separating and purifying multiple types of cells and in separating magnetic beads from sorted cells. In addition, magnetic bead bonding may damage cell membrane proteins and structure. The damaged cells are inconducive for subsequent cell culture.

### 2.6. Micro-Robot-Assisted Method

Recently, with the improvement of MEMS and NEMS technology, various micro and nanoscale microstructures, which are named microrobots, have been fabricated. In most applications, they can act as microcarrier to delivery drugs or cells in vitro or in vivo [83]. And different driven mechanisms have been developed to control microrobot. Commonly, the propulsion mechanisms of these microrobots can be divided into three categories: Chemical means, physical means, and biological means [84]. In other cases, microrobot can also be used to manipulate single-cell for its micro-scale size. Additionally, microfluidic chip can be combined for it can not only provide a simulated vascular environment or a microscale chamber, but also have high throughput and have high repeatability [83]. Feng et al. fabricated silicon-based microrobot and proposed acoustic levitation driven method to 3-D rotate single oocyte in a microfluidic [39]. The positioning accuracy is less than 1 μm and orientation with an accuracy of one and an average rotation velocity of 3 rad/s were achieved. Moreover, they used a microrobot in microfluidic chip to transport cell in contactless manner, as shown in Figure 7c [82]. A local vortex can be generated after oscillating the microrobot by permanent magnets. Different streamline can be generated when various oscillation amplitudes, frequencies, and relative positions between microrobot and microchannel wall are adapted. The tuning of these parameters changes the viscous fluid dynamics and cells can be transported. Using microrobot on a chip to manipulate single cells is promising owing to the microrobot’s high accuracy, high speed, non-contact, enough physical strength, applicability to different sizes cells, and microfluidic chip’s sealing property, and also high throughput.

## 3. Microfluidic Single-Cell Analysis

Various analytical functionalities, including microscopy, microelectrodes array, mass spectrometry, and chromatography, can be integrated with microfluidic components for various qualitative and quantitative single-cell analysis applications [85]. Fluorescence microscopy is the most widely used microfluidic technique for cell analysis [86]. Given the good optical transparency of microfluidic chips, various types of microscopy techniques can be integrated to image the morphology, structure, and migration of cells and specifically labeled subcellular organelles. Optical detectors for absorbance, laser-induce fluorescence (LIF), and chemiluminescence (CL) can also be integrated with microfluidic channels for detecting and quantifying specific biomolecules [87]. Electrical analysis, such as electrochemical impedance spectroscopy (EIS) and patch-clamp, can also be incorporated into microdevices to monitor cell secretion, morphology, and migration [88,89]. Mass spectrometry is a powerful analytical technique that has been coupled with microfluidics for the analysis of cellular contents (DNA, proteins, and glycan) and metabolites [90]. The integration of microfluidic systems with these analytical approaches enables rapid, sensitive, reproducible, and high-throughput single-cell analysis, which promotes the development of basic biological studies and clinical diagnoses and therapies [91]. This section provides review of microfluidics based single-cell analysis by using integrated analytical techniques.

### 3.1. Cellular Analysis

The study of cellular behavior, such as cell morphology, migration, proliferation, differentiation, and apoptosis, has general scientific and practical value for biology and medicine. However, several important cellular behaviors occur in in vivo environments that cannot be easily implanted with sensors or other types of molecular probes. Thus, an alternative approach is necessary to transfer the cells of interest out of their natural environment to one that is more conducive for the measurement scheme. The disadvantage of this alternative approach is that cell behavior in vitro may be different from that in vivo. In addition, many of the stimuli to which the sample is subjected to in vivo are no longer present in vitro. Microfluidic chips that integrate chemical, mechanical, and electrical functionalities in a lab-on-a-chip system may be powerful tools for mimicking in vivo microenvironments for cell growth. Moreover, many of the materials used to construct the chips are optically transparent, making them ideal for the real-time monitoring of cellular behavior through imaging. Therefore, microfluidics is suitable for cellular behavior analysis, and different microfluidic platforms have been developed for various applications.

#### 3.1.1. Morphology

Cells are surrounded by a myriad of physical and biochemical cues in a cellular microenvironment. Cell morphology is the most intuitionistic parameter that can reflect cellular responses to different stimuli. The quantitative morphological analysis of cells is a key approach for abnormality identification and classification, early cancer detection, and dynamic change analysis under specific environmental stress. Quantitative results guide pathologists in making final diagnostic decisions. By integrating biomimetic cell culture systems with various types of microscopy or electrical techniques, microfluidics offers a robust platform for the real-time monitoring of alterations in cell morphology.

Fluorescence imaging is the most commonly used technique for cell morphology observation. In this approach, cells are cultured in microchannels and labeled with fluorescent dyes or proteins for visualization under fluorescence microscopy. Sung Ke et al. developed simple straight channel arrays as a viable and robust tool for the high-throughput quantitative morphological analysis of single MSCs and the examination of cell–material interactions [92]. Wu et al. developed a novel microfluidic model and studied the influences of interstitial flows on cell morphology. They found that interstitial flows promote amoeboid cell morphology and motility of MDA-MB-231 cells [93]. As shown in Figure 8a, Qin et al. have investigated particular longevity-related changes in cell morphology and characteristics of yeast cells in a microfluidic single-cell analysis chip. They found that cells with the round-budded terminal morphology had longer lifespans than those with the elongated-budded morphology [94]. Electrical techniques can also be incorporated in microfluidic devices for cell morphology analysis. Andreas Hierlemann et al. developed a microfluidic single-cell impedance cytometer that can perform the dielectric characterization of single-cells under frequencies of up to 500 MHz [95]. The increase in working frequency enabled the characterization of subcellular features in addition to the properties that are visible at low frequencies. The capabilities of this electrical cytometer have been demonstrated in the discrimination of a wild-type yeast strain from a mutant strain based on differences in vacuolar size and intracellular fluid distribution. One year later, Andreas Hierlemann et al. reported a microfluidics-based system that can reliably capture single rod-shaped Schizosaccharomyces pombe cells by applying suction through orifices in a channel wall. This system enables the subsequent culturing of immobilized cells in an upright position. Dynamic changes in cell cycle state and morphology are continuously monitored through EIS over a broad frequency range [96]. The obtained results showed that the spatial resolution of the measured cell length is 0.25 μm, which corresponds to a 5 min interval of cell growth under standard conditions. Comprehensive impedance datasets have also been used to determine the occurrence of nuclear division and cytokinesis.

#### 3.1.2. Proliferation

Cell proliferation is the process through which the number of cells increases. It is carefully balanced with cell death to maintain a constant number of cells in adult tissues and organs. Cell proliferation analyses are crucial for cell growth and differentiation studies, and they are generally used to evaluate the toxicity of compounds and the inhibition of tumor cell growth during drug development. Microfluidic technology provides precise, controlled, cost-effective, compact, integrated, and high-throughput microsystems that are promising substitutes for conventional biological laboratory methods for the study of single-cell proliferation. Microfluidics allows dynamic cell culture in micro perfusion systems to deliver continuous nutrient supplies for long-term cell culture. In addition, this strategy offers many opportunities for mimicking the cell–cell and cell–extracellular matrix interactions of tissues by creating gradient concentrations of biochemical signals, such as growth factors, chemokines, and hormones. Many applications of cell cultivation in microfluidic systems are aimed toward understanding the proliferation and differentiation of cell populations.

The analyses of clonal cultures established from single-cells are vital for cancer research because heterogeneity plays an important role in tumor formation. Microfluidics-based devices are the most ideal choice for high-throughput single-cell clonal expansion. As shown in Figure 8b, Carl L Hansen et al. presented a simple microfluidic cell-culture design that supports cell growth and replicates standard microcultures [97]. Culture conditions can be precisely controlled on the microfluidic chip, which can also be applied for the in-situ immunostaining and recovery of viable cells. The platform successfully mimics conventional cultures in reproducing the responses of various types of primitive mouse hematopoietic cells, while retaining their functional properties, as demonstrated by the subsequent in vitro and in vivo (transplantation) assays of the recovered cells. Justin Cooper-White et al. reported a two-layered microfluidic device platform for the capture, culture, and clonal expansion of single-cells [35]. Under the manual injection of a cell suspension, hundreds to thousands of single-cells (adherent and nonadherent) are deterministically trapped in a high-throughput manner, and high trapping efficiency is achieved by incorporating a U-shaped hydrodynamic trap in the downstream wall of each microwell. They confirmed that the modified microwells promote the attachment, dispersal, and proliferation of the trapped single-cells for multiple generations over extended periods of time (>7 days) under media perfusion. Chia-Hsien Hsu et al. proposed a microfluidic device with a dual-well (DW) design for high-yielding single-cell loading (~77%) in large microwells (285 and 485 µm in diameter). This device facilitates cell dispersal, proliferation, and differentiation [35]. The architecture of this device allows the size of the “culture” microwells to be flexibly adjusted without affecting single-cell loading efficiency, making it useful for cell culture applications, as demonstrated by their experiments on KT98 mouse neural stem-cell differentiation, A549 and MDA-MB-435 cancer cell proliferation, and A549 single-cell colony formation.

#### 3.1.3. Migration

Cell migration refers to the movement of cells in response to biological signals and environmental cues. This process plays a vital role in key physiological processes, including immune cell recruitment, wound healing, tissue repair, embryonic morphogenesis, and cancer metastasis. The complex processes that govern cell migration must be comprehensively understood to promote the development of novel therapeutic strategies. Cell migration is regulated by several biological, chemical, and physical signals, including mechanotransduction, chemical signaling, and molecular interactions. Given that cell migration is a highly complex biological mechanism, it can only be elucidated through monitoring under defined physiologically conditions. However, in vivo cell migration studies using state-of-the-art imaging methods are hindered by ethical issues associated with animal testing. Moreover, the tracking of cell migration in vivo remains technically challenging. Thus, in vitro migration assays are extensively used by biologists, pharmacologists, medical researchers, and toxicologists for diverse applications. The traditional scratch assay, which is the most convenient and inexpensive method for in vitro cell migration analysis, has several limitations. For example, the 2D scratch assay cannot replicate the 3D environment of cells and the signal gradients that are present in vivo. In addition, this method precludes single-cell analysis and cannot reveal cell heterogeneity, which is a vital factor of cancer metastasis. Microfluidics has emerged as a powerful platform for the study of cancer migration given that it can provide well-defined environmental cues. Microfluidic devices require a low number of cells and are highly suitable for high-throughput single-cell screening. As shown in Figure 8c, Euisik Yoon et al. developed a single-cell migration platform that allows the examination of the migration behavior of individual cells and the sorting of a heterogeneous cell population on the basis of chemotactic phenotype [98]. Highly chemotactic and nonchemotactic cells have been retrieved for the further cellular and molecular analyses of their differences. The migration channel has also been modified to elucidate the movement of certain cancer cells through geometrically confined spaces.

#### 3.1.4. Apoptosis

Programmed cell death, which is known as apoptosis, is a vital component of various processes, including normal cell turnover, proper immune system development and function, hormone-dependent atrophy, embryonic development, and chemical-induced cell death. Inappropriate apoptosis (either inadequate or excessive) is a factor of many human conditions, including neurodegenerative diseases, ischemic damage, autoimmune disorders, and many cancer types. At present, the study of apoptosis is progressing rapidly.

Current methods for evaluating the effects of agents against cell apoptosis are generally expensive, labor-intensive and heterogeneity-ignored because they involve the use of multi-well plates that are operated using cumbersome manual or expensive robotics-based operations to evaluate the average results of a population. Therefore, researchers must urgently develop a technology that can perform such experiments in a cheaper, easier, and higher throughput manner to analyze cell apoptosis at the single-cell level. Microfluidic chips present the advantages of ease of integration and the potential for high-throughput single-cell manipulation, making them attractive platforms for drug metabolism and cell cytotoxicity analyses. As shown in Figure 8d, Wlodkowic et al. used a microfluidic single-cell array chip for the real-time analysis of events leading up to apoptosis in model cell lines [19]. They found that these live-cell, microfluidic microarrays can be readily applied to kinetic analysis of investigational anticancer agents in hematopoietic cancer cells, providing new opportunities for automated microarray cytometry and higher-throughput screening. Through quantifying the anticancer drug induced apoptosis on-chip, they showed that, with small numbers of trapped cells (∼300) under careful serial observation, they can achieve results with only slightly greater statistical spread than that can be obtained with single-pass flow cytometer measurements of 15,000–30,000 cells. Kumar et al. used digital microfluidics (DMF) for time-resolved cytotoxicity studies on single non-adherent yeast cells [100]. They achieved real-time monitoring of single yeast cell responses during antifungal treatment in a high-throughput manner, and their DMF platform with microwell arrays is demonstrated as a promising tool for implementing various biological applications concerning single non-adherent cells in a high-throughput manner. Li et al. developed a multifunctional gradients-customizing microfluidic device for high-throughput single-cell multidrug resistance (MDR) analysis [101]. Bithi and Vanapalli reported a microfluidic cell isolation technology for drug testing of single tumor cells and their clusters, and they found that individual tumor cells display diverse uptake profiles of the drug [102]. Experiments with clusters of tumor cells compartmentalized in their microfluidic drops revealed that cells within a cluster have higher viability than their single-cell counterparts when exposed to doxorubicin.

#### 3.1.5. Differentiation

Stem cells can continuously self-renew and have the potential to differentiate into specific tissues. Thus, their roles in tissue engineering, organ regeneration, cell-based therapies, disease models, drug development, and various healthcare applications have been extensively investigated for over 50 years. To date, stem cells have been successfully used to heal damaged tissues and replace nonfunctional organs. The promising applications of stem cells in the biological and therapeutic fields have been hindered by the challenges associated with the maintenance of undifferentiated pluripotency and the reliable direction of stem-cell differentiation. Conventional cell culture methods, such as those based on petri dishes or Transwells, cannot achieve an in vivo-like microenvironment that contains diverse well-controlled stimuli. The emergence and rapid development of microfluidics have presented a possible solution for mimicking an in vivo-like microenvironment. Microfluidic platforms can precisely manipulate the microenvironment to deliver soluble factors to cells, establish well-defined gradients, integrate various biocompatible scaffolds and functional components, and dynamically alter the application of mechanical and electrical signals to cultured cells. The combination of microfluidic technologies with stem-cell analysis could finally provide in-depth insight into stem-cell differentiation mechanisms to enable their application. At present, an increasing number of works have focused on applying microfluidic devices to investigate stem-cell differentiation at single-cell resolution.

Sikorski et al. developed a microfluidic device that supports the robust generation of colonies derived from single human embryonic stem cells (hESCs) [103]. The use of this device to analyze the clonal growth of CAIS hESCs demonstrated its ability to reveal the heterogeneity of differentiation patterns displayed by clonally tracked hESC. In addition to providing controllable microenvironments for directing and observing stem-cell differentiation, microfluidics can also be used to characterize differentiation status. As shown in Figure 8e, Zhou et al. designed and fabricated a microfluidic device that integrates the hydrodynamic trapping of single-cells in predefined locations with the capability to perform electrical impedance measurements [99]. Mouse embryonic stem cells at different states during differentiation (t = 0, 24, and 48 h) were measured and quantitatively analyzed. The magnitude of cell impedance markedly increased. This increase can be attributed to the increase in cell size. The analysis of the measurements suggested that the nucleus-to-cytoplasm ratio decreased during this process. The maximum degree of cell heterogeneity was observed when the cells were in the transition state (24 h).

#### 3.1.6. Metabolism

The intracellular levels and spatial localizations of metabolites reflect the state of a cell and its relationship to its surrounding environment [104]. Microfluidic device is an ideal platform for cellular metabolite profiling both in physiological environment and under drug treatment, owing to the ability of integrating cell culture, stimulation, metabolite enrichment, and detection on a single chip coupled with various analytical instruments [105].

Among diverse analytical techniques, Mass Spectrometry is the most powerful and promising tool for cell metabolite analysis, because of its broad detection range, high sensitivity, high mass resolution, rapid operation, and the ability for multiplexed analysis. Zhang et al. integrated droplet-based microfluidics with mass spectrometry for high-throughput and multiple analysis of single-cells [106]. Specific extraction solvent was used to selectively obtain intracellular components of interest and remove interference of other components. Using this method, matrix-free, selective, and sensitive detection of metabolites in single-cells is easily realized. Optical detecting techniques can also be integrated with microfluidic device for cell metabolite analysis. Wang et al. presented a flexible high-throughput approach that used microfluidics to compartmentalize individual cells for growth and analysis in monodisperse nanoliter aqueous droplets surrounded by an immiscible fluorinated oil phase [107]. The fluorescent assay system was used to measure the concentration of the metabolites (oxidase enzymes), and the assay reaction started when a cell-containing droplet coalesced with an assay droplet. As shown in Figure 8f, Ben et al. proposed a label-free method for exploiting the abnormal metabolic behavior of cancer cells. A single-cell analysis technique is used to measure the secretion of acid from individual living tumor cells compartmentalized in monodisperse, picolitre (pL) droplets. As few as 10 tumor cells can be detected in a background of 200,000 white blood cells and proof-of-concept data was shown on the detection of CTCs in the blood of metastatic patients.

### 3.2. Genetic Analysis

Genetic analysis is one of the most important and extensively developed field in microfluidic single-cell analysis. Genetic analysis can be categorized into cytogenetic and molecular genetic analysis.

#### 3.2.1. Cytogenetic Analysis

Cytogenetic analysis deals with chromosomes and related abnormalities and is very crucial in the diagnosis of oncologic and hematologic disorders. The methods for cytogenetic analysis usually include Karyotyping and fluorescence in-situ hybridization (FISH). Karyotyping helps detect structural or numerical chromosome abnormalities. Chromosome analyses require cell cultures and involve the harvesting of chromosomes, chromosome banding, microscopic analysis, and the production of karyotypes. FISH involves the determination of the presence, absence, position, and copy number of DNA segments with the help of fluorescence microscopy. The most popular cytogenetic analysis on microfluidic chips is based on the using FISH. Shah et al. described a novel microfluidic FISH preparation device for metaphase FISH slides preparation [108]. The device combines the bioreactor for cell culturing with the splashing device for preparation of the chromosome spreads. As shown in Figure 9a, Sieben et al. have successfully integrated all aspects needed to perform automated FISH on a microfluidic platform [109]. They detected the number of X and Y chromosomes per cell in patient samples; useful for identifying the status of engraftment in patient-donor sex-mismatched transplantation. Zanardi et al. presented a microfluidic-device-based FISH method performed on fresh and fixed hematological samples, which integrated cluster-assembled nanostructured TiO2 (ns-TiO2) as a nanomaterial promoting hematopoietic cell immobilization in conditions of shear stress. By this way, FISH can be performed with at least a 10-fold reduction in probe usage and minimal cell requirements, but had comparable performance to standard FISH, indicating that it is suitable for genetic screenings in research clinical settings.

#### 3.2.2. Molecular Genetic Analysis

Molecular genetic analysis studies the structure and function of genes at a molecular level and thus employs methods of both molecular biology and genetics. Nucleic acid amplification processes play a critical role in sensitive detection and quantification, because the amount of the nucleic acids extracted from cells is small. Polymerase chain reaction (PCR) is the most widely used non-isothermal amplification technique, which performs thermal cycling to amplify a particular DNA sequence to generate thousands to millions of copies. Quantification of RNA can be achieved by performing reverse-transcription PCR (RT-PCR). As for genetic analysis, microfluidic devices have advantages including faster reaction times, low sample consumption, precise temperature distribution and the ease of integrating with separation techniques.

Even genetically identical cells with seemingly identical cell histories and environmental conditions can have significant differences in gene expression levels, due largely to the alteration of mRNA production by random fluctuations or complex molecular switches. Thus, quantitative analysis of gene expression at single-cell level is important for the understanding of basic biological mechanism and disease onset and progression [111]. Currently, several microfluidic-based single-cell RNA-Seq platforms have been developed and applied to study transcriptional heterogeneity of cancer, immune [112], and stem cells [29]. Those microfluidic-based single-cell RNA-Seq platforms are basically based on either active-valve or droplet-based microfluidics. As shown in Figure 9b, White presented a valve-based fully integrated microfluidic device capable of performing high-precision RT-qPCR measurements of gene expression from hundreds of single-cells per run [110]. They applied this technology to 3,300 single-cell measurements of miRNA expression in K562 cells, coregulation of a miRNA and one of its target transcripts during differentiation in embryonic stem cells, and single nucleotide variant detection in primary lobular breast cancer cells. Huang et al. also developed a valve-based strategy for single-cell RNA-Seq that has superior sensitivity and been implemented in a microfluidic platform for single-cell whole-transcriptome analysis [113]. In their approach, single-cells were captured and lysed in a microfluidic device, where mRNAs with poly(A) tails were reverse-transcribed into cDNA. Double-stranded cDNA was then collected and sequenced using a next generation sequencing platform. Droplet microfluidics is among the most promising candidate for capturing and processing thousands of individual cells for whole-transcriptome or genomic analysis in a massively parallel manner with minimal reagent use. As shown in Figure 9c, Klein et al. recently established a method called in Drops, which is based on the using of droplet microfluidics and has the capability to index >15,000 cells in an hour [30]. A suspension of cells was first encapsulated into nanoliter droplets with hydrogel beads (HBs) bearing barcoding DNA primers. Cells were then lysed, and mRNA is barcoded (indexed) by a reverse transcription (RT) reaction.

### 3.3. Protein Analysis

Proteins are one basic component of cells, which perform and regulate various cellular functions. Owing to the low abundance and high complexity, the development of sensitive and reliable protein analysis techniques is highly desirable. Microfluidics offer rapid, sensitive, reproducible and high-throughput platforms for protein analysis. Various aspects, including protein species, amounts, activity, as well as protein interaction with other biomolecules, can be analyzed using microfluidic devices, with tremendous advantages over conventional methods [114].

Cellular staining assays are commonly used methods that are easy to be applied in microfluidic devices for protein analysis. Proteins in cells are specifically labeled by tags or fluorescent antibodies, and their locations and expressions can be imaged using microscopies. As shown in Figure 10a, Srivastava et al. reported a novel phosphoFlow Chip (pFC) that relies on monolithic microfluidic technology to rapidly conduct signaling studies. The pFC platform integrates cell stimulation and preparation, microscopy, and subsequent flow cytometry [115]. Except for intracellular protein analysis, microfluidics can also be used for the analysis of secreted proteins of single-cells. As shown in Figure 10b, Ma et al. reported a microfluidic platform designed for highly multiplexed (more than ten proteins), reliable, sample- efficient (~1 × 10^4^ cells) and quantitative measurements of secreted proteins from single-cells [116]. They validated the platform by assessment of multiple inflammatory cytokines from lipopolysaccharide-stimulated human macrophages and comparison to standard immune-technologies. Another important microfluidic protein analysis technique is surface-based immunoassay. Proteins are specifically captured by affinity ligands modified on microchannel or microbead surface, and sandwich immunoassays are then performed. Godwin et al. reported an integrated microfluidic approach that enables on-chip immune-isolation and in situ protein analysis of exosomes directly from patient plasma [117]. Specifically, a cascading microfluidic circuit was designed to streamline and expedite the pipeline for proteomic characterization of circulating exosomes, including exosome isolation and enrichment, on-line chemical lysis, protein immunoprecipitation, and sandwich immunoassays assisted by chemi-fluorescence detection. This method enables high level of multiplexing and quantitation, and intracellular, membrane, and secreted proteins can all be analyzed from the same single-cell [114]. Recently, protein immunoblotting assay has been operated on microfluidic devices, and microfluidic single-cell Western blotting (scWestern) has also been developed [44]. Polyacrylamide gels were photo-patterned to form a microwell array, in which single-cells were settled and lysed in situ. Gel electrophoresis was then performed, and separated proteins were immobilized by photoinitiated blotting and detected by antibody probing. This scWestern method enabled multiplexed analysis of 11 protein targets per single-cell with detection thresholds of <30,000 molecules.

### 3.4. Biophysical Property Analysis

Cell state is often characterized through measurement of biochemical and biophysical markers. Although biochemical markers have been widely used, intrinsic biophysical markers, such as size, density, and the ability to mechanically deform under a load, are advantageous in that they do not require costly labeling or sample preparation [41]. Great cellular heterogeneity also exists in these biophysical properties, and microfluidics is an ideal technology for analyzing different biophysical properties at the single-cell level.

As shown in Figure 11a, Godin et al. developed a suspended microchannel resonator (SMR) combined with picolitre-scale microfluidic control to measure buoyant mass and determine the ‘instantaneous’ growth rates of individual cells [118]. The SMR measures mass with femtogram precision, allowing rapid determination of the growth rate in a fraction of a complete cell cycle. They found that for individual cells of Bacillus subtilis, Escherichia coli, Saccharomyces cerevisiae and mouse lymphoblasts, heavier cells grew faster than lighter cells. Not only the mass, but also the size and density of the cells can be measured by using SMR [119,120,121]. The mechanical property of single-cells can be used to evaluate the status of many diseases including cancer, malaria, and arthritis. Microfluidics is a powerful technology for characterizing the mechanical properties of single-cells at a fast and high-throughput manner. Guo et al. developed a microfluidic chip for measuring the deformability of single-cells using the pressure required to deform such cells through micrometer-scale tapered constrictions [122]. Single-cells are infused into a microfluidic channel, and then deformed through a series of funnel-shaped constrictions. The constriction openings are sized to create a temporary seal with each cell as it passes through the constriction, replicating the interaction with the orifice of a micropipette. They measured the deformability of several types of nucleated cells and determined the optimal range of constriction openings. Hu et al. developed a microfluidic elasticity microcytometer for multiparametric biomechanical phenotypic profiling of live single cancer cells for quantitative, simultaneous characterizations of cell size, and cell deformability/stiffness [123]. The elasticity microcytometer was implemented for measuring and comparing four human cell lines with distinct metastatic potentials. Except for using passive hydrodynamic pressure for cell deformability, active methods such as optical tweezer and DEP can also be combined with microfluidic chip for studying biomechanics of single-cells. For instance, Zhang et al. developed a microfluidic chip for rapid characterization of the biomechanics of drug-treated cells through stretching with dielectrophoresis (DEP) force, and saw a decrease in the stiffness after drug treatment of NB4 cells [124]. Electrical impedance is also an important biophysical marker for label-free identification of different cell types or detecting intracellular changes [125]. Microfluidics has great capability in measuring electrical impedance of single-cells because microelectrode array can be easily integrated into microfluidic chips. Hong et al. proposed a method for differentiating four kinds of cell (HeLa, A549, MCF-7, and MDA-MB-231) using impedance measurements at various voltages and frequencies [126]. According to the impedance measurements, HeLa, A549, and MCF-7 cells and the pathological stages of a given cancer cell line (MCF-7 and MDA-MB-231) can be distinguished. Measuring two or more than two biophysical markers for the same single-cells leads to more comprehensive understanding of the linkage between biological and biophysical things. As shown in Figure 11b, Zhou et al. developed a microfluidic device that can simultaneously characterize the mechanical and electrical properties of individual biological cells in a high-throughput manner (>1000 cells/min) [127]. The combination of mechanical and electrical properties provides better differentiation of cellular phenotypes, which are not easily discernible via single biophysical marker analysis.

## 4. Discussion and Conclusions

The use of microfluidic technology for diversified and efficient manipulation and analysis of single biological cells has been a research hotspot in the interdisciplinary field. This review highlighted the microfluidic single-cell manipulation and analysis from the aspects of methods and applications. Specifically, the microfluidic single-cell manipulation can be flexibly realized by using hydrodynamic, electrical, optical, magnetic, and acoustic and micro-robot assisted methods, and microfluidic chips can be combined with various analytical techniques for single-cell analysis ranging from cellular behaviors to secreted proteins. The advantages and disadvantages of some methods are discussed. It is seen that each method has its inherent advantages and disadvantages (Table 2). In general, hydrodynamic method can achieve high throughput manipulation of cell samples, but there are deficiencies in the accuracy and flexibility. Methods such as electrical and optical methods have high accuracy and great flexibility, but they have shortcomings such as low throughput. There is no single method that can fulfill high throughput, high-efficiency, accurate single-cell manipulation and analysis tasks simultaneously. Therefore, to meet the specific requirements of practical applications, multiple methods are integrated. Secondly, existing technologies should be continuously improved for better single-cell manipulation and analysis. For example, the microstructure can be extended from 2-dimensional to 3-dimensional and fabrication crafts with higher precise can also be developed. Thirdly, new mechanisms and technologies must be discovered. The research can be based on theoretical simulations revealing the fundamental theories involved in microfluidics. Fourthly, developing a precise fluidic control system with fast response is very important. For instance, Arai et al. proposed a high-speed local-flow control using dual membrane pumps driven by piezoelectric actuators placed on the outside of microfluidic chip, and their approach can sort single cells at throughput of 23,000 cells/s with a 92.8% success rate, 95.8% purity, and 90.8% cell viability [128]. Finally, microfluidic chip implanted in subcutaneous, blood vessels and other tissues and organs, for achieving precise single-cell manipulation and analysis directly in the human body, is an emerging direction.

## Figures and Tables

**Figure 1 micromachines-10-00104-f001:**
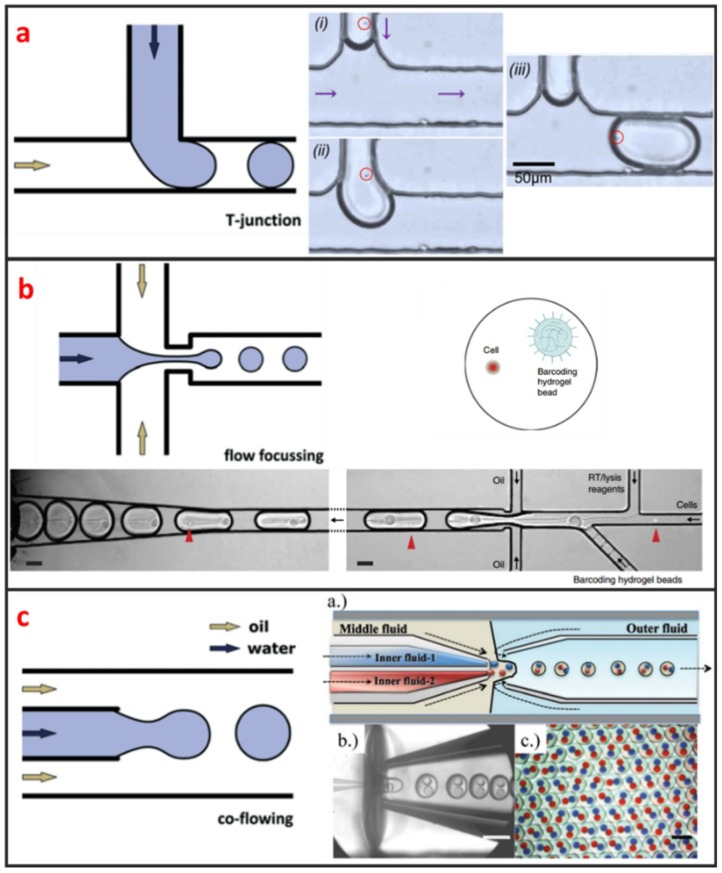
Three types of methods and designs for microfluidic single-cell droplet generation. Adapted by permission from Reference [28], copyright Royal Society of Chemistry 2015. (**a**) T-junction microchannel based droplet generation for encapsulating single cells for cultivation and genomic analysis. Adapted by permission from reference [49], under the Creative Commons Attribution License 2017. (**b**) Flow-focusing for encapsulating a single cell and a barcoding hydrogel bead in a droplet for single-cell barcoding and sequencing. Adapted by permission from Reference [27], copyright Nature Publishing Group 2017. (**c**) Co-flow for preparing multiple component double emulsions. Adapted by permission from Reference [50], copyright Royal Society of Chemistry 2012.

**Figure 2 micromachines-10-00104-f002:**
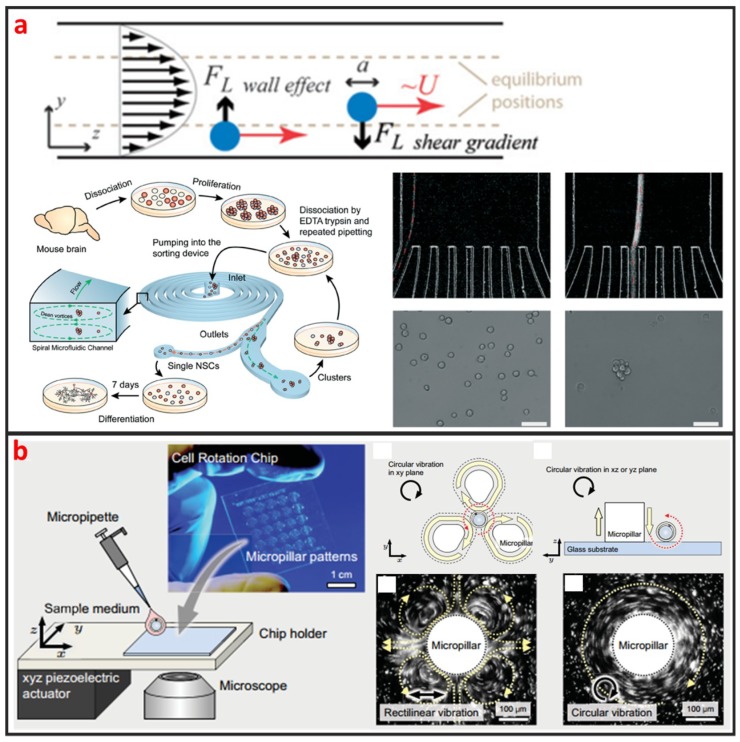
Inertial and vortex microfluidic single-cell manipulation and some designs. (**a**) Inertial microchannel to separate single-cell from cell cluster. Adapted by permission from References [52,55], copyright Royal Society of Chemistry 2009 and 2015. (**b**) Vortex generated by micropillar to rotate single-cell. Adapted by permission from Reference [58], under the Creative Commons Attribution License 2015.

**Figure 3 micromachines-10-00104-f003:**
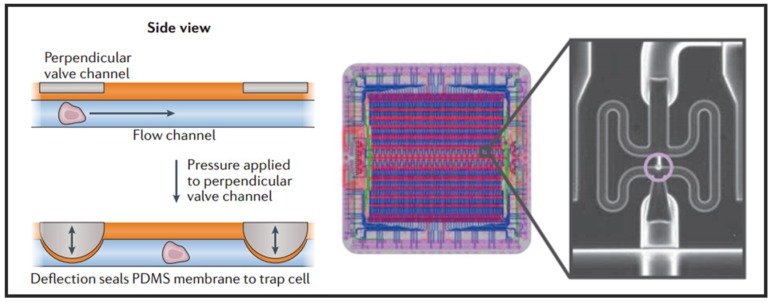
Microfluidic valve single-cell isolation and a design Adapted by permission from References [29,60], copyright Nature Publishing Group 2017 and 2014.

**Figure 4 micromachines-10-00104-f004:**
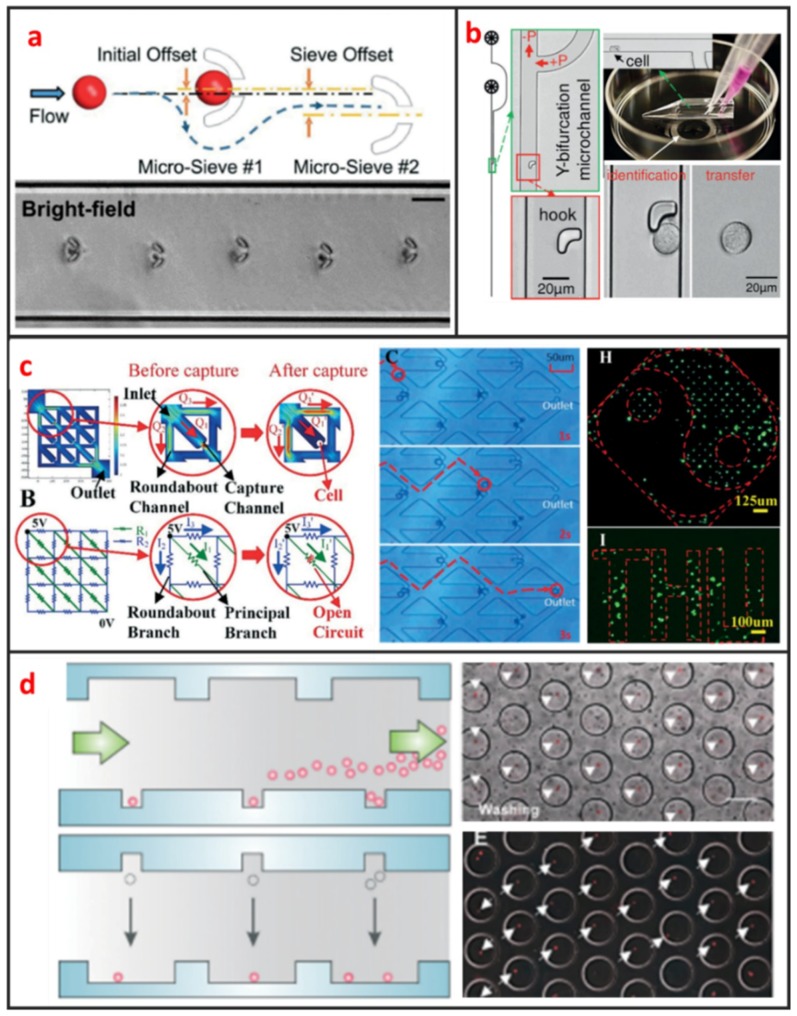
Microstructure based single-cell manipulation methods and some typical designs. (**a**) Micro-sieve to isolate floating single cancer cell under continuous flow. Adapted by permission from Reference [62], copyright Royal Society of Chemistry 2016. (**b**) A microfluidic pipette tip with a micro-hook for trapping and releasing a single cell. Adapted by permission from Reference [37], copyright Royal Society of Chemistry 2016. (**c**) Microchannel to trap single-cell based on fluidic circuit. Adapted by permission from Reference [10], copyright Royal Society of Chemistry 2016. (**d**) A dual-microwell design for the trap and culture of single cells. Adapted by permission from Reference [66], copyright Royal Society of Chemistry 2015.

**Figure 5 micromachines-10-00104-f005:**
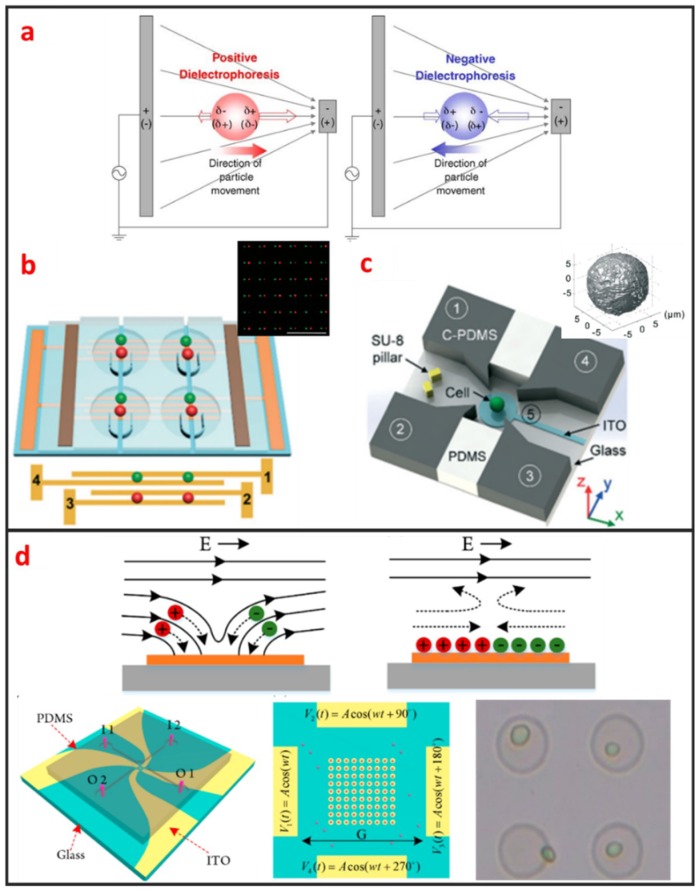
Electrical single-cell manipulation methods and some designs. (**a**) Theory of dielectrophoresis (DEP). Adapted by permission from Reference [68], copyright Elsevier 2005. (**b**) 2D electrode to trap and pair single cells. Adapted by permission from Reference [20], copyright Royal Society of Chemistry 2017. (**c**) 3D electrode to rotate single cell. Adapted by permission from Reference [69], copyright Royal Society of Chemistry 2018. (**d**) Rotating electric field induced-charge Electro-osmosis to trap single cell. Adapted by permission from Reference [70], copyright American Chemical Society 2016.

**Figure 6 micromachines-10-00104-f006:**
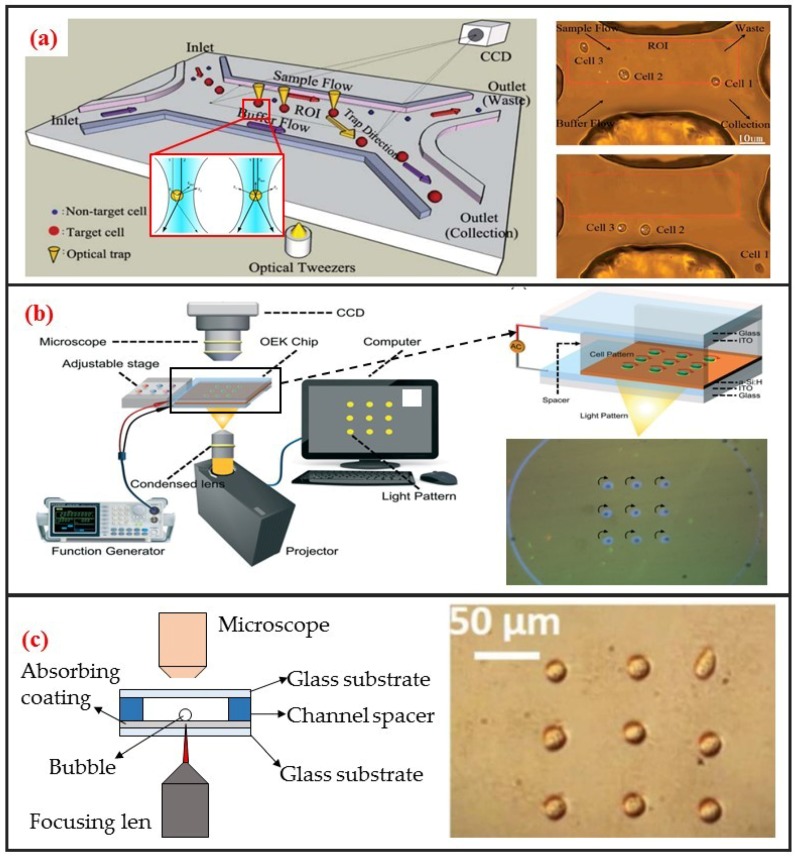
Optical single-cell manipulation methods and some designs. (**a**) Optical tweezer based single-cell sorting. Adapted by permission from Reference [12], copyright Royal Society of Chemistry 2011. (**b**) ODEP based single-cell array rotation. Adapted by permission from Reference [76], copyright Royal Society of Chemistry 2017. (**c**) Opto-thermocapillary based single-cell pattern. Adapted by permission from Reference [77], copyright Royal Society of Chemistry 2013.

**Figure 7 micromachines-10-00104-f007:**
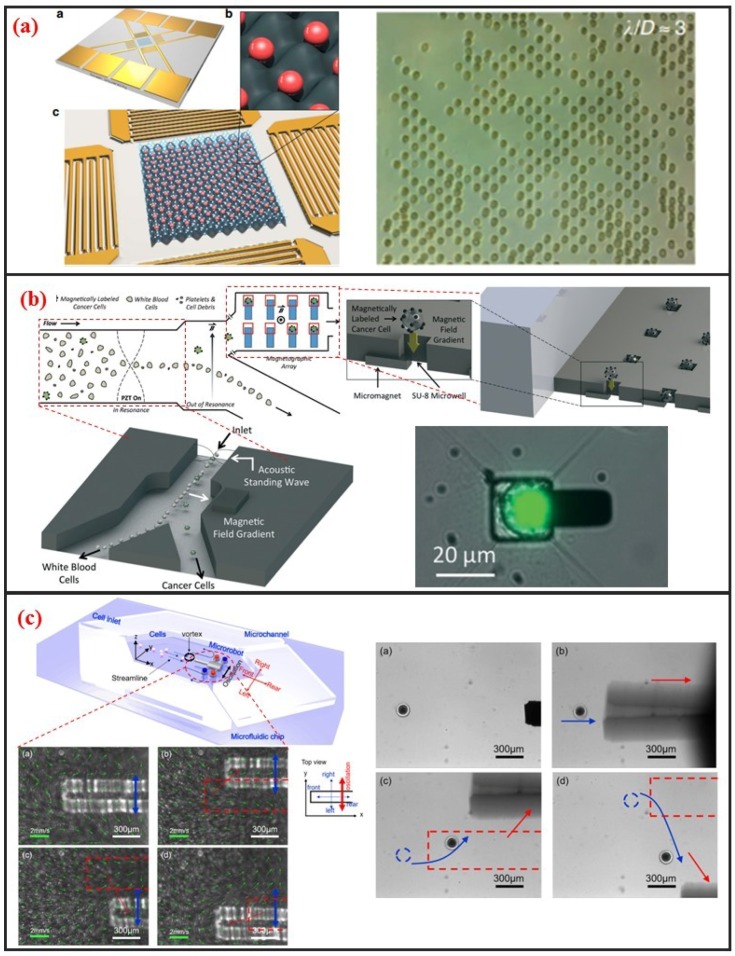
Acoustic, magnetic and microrobot single-cell manipulation methods and some designs. (**a**) Surface acoustic waves based single-cell pattern. Adapted by permission from Reference [23], copyright Nature Publishing Group 2015. **(b**) Micromagnet based single-cell trap. Adapted by permission from Reference [81], copyright Royal Society of Chemistry 2016. (**c**) Noncontact cell transportation by oscillation of microrobot in microfluidic chip. Adapted by permission from Reference [82], copyright AIP Publishing 2017.

**Figure 8 micromachines-10-00104-f008:**
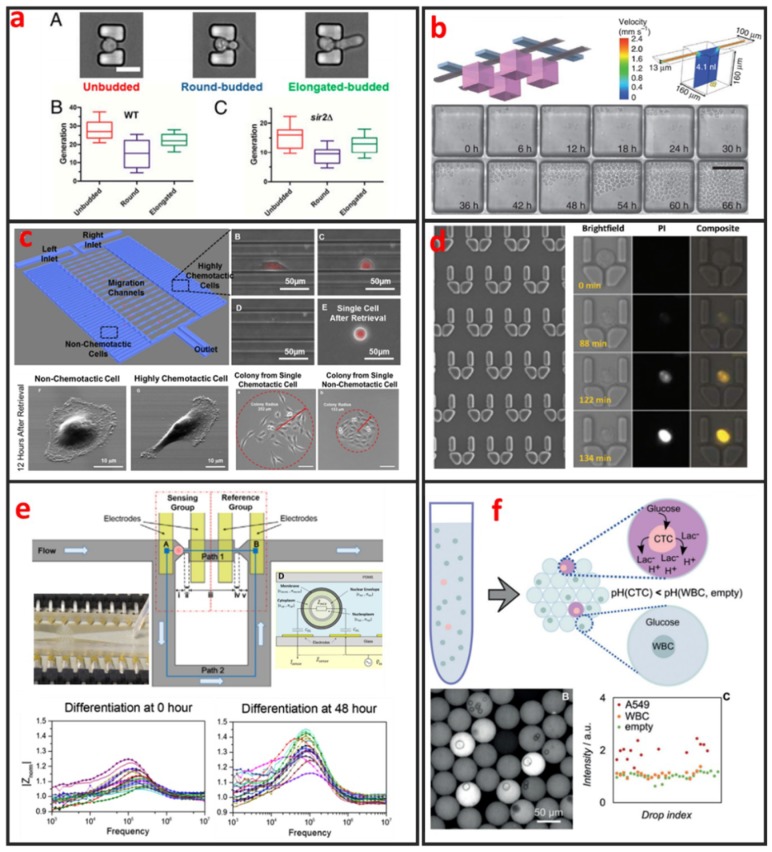
Cellular analysis of single-cells by using microfluidics. (**a**) Characterization of terminal morphology in aging yeast cells. Adapted by permission from Reference [94], copyright National Academy Sciences 2015. (**b**) High-throughput analysis of single hematopoietic stem cell proliferation in microfluidic cell culture arrays. Adapted by permission from Reference [97], copyright Nature Publishing Group 2011. (**c**) Single-cell migration chip for chemotaxis-based microfluidic selection of heterogeneous cell populations. Adapted by permission from Reference [98], under the Creative Commons Attribution License 2015. (**d**) Dynamic analysis of drug-induced cytotoxicity using microfluidic single-cell array. Adapted by permission from Reference [19], copyright American Chemical Society 2009. (**e**) Single-cell studies of mouse embryonic stem cell (mESC) differentiation by electrical impedance measurements in a microfluidic device. Adapted by permission from Reference [99], under the Creative Commons Attribution License 2016. (**f**) Circulating tumor cells (CTCs) detection based on the Warburg effect using single-cell compartmentalization in microdroplets. Adapted by permission from Reference [26], copyright John Wiley and Sons 2016.

**Figure 9 micromachines-10-00104-f009:**
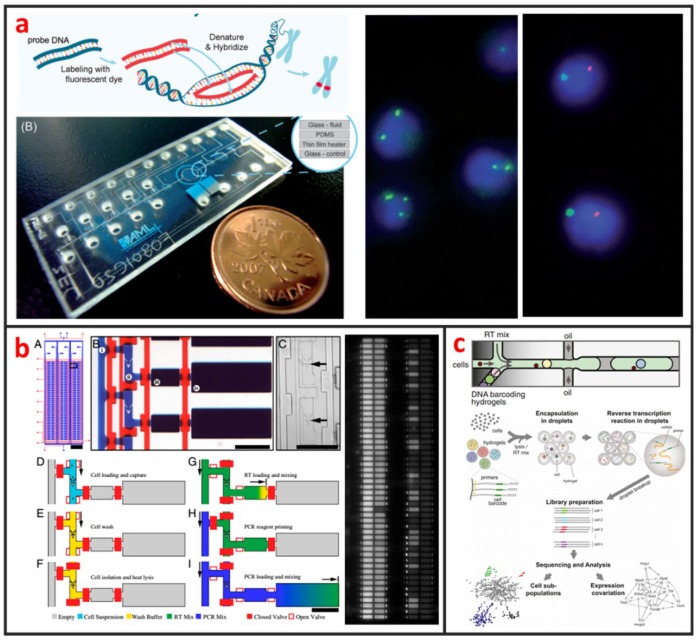
Genetic analysis of single-cells by using microfluidics. (**a**) An integrated microfluidic chip for chromosome enumeration using FISH. Adapted by permission from Reference [109], copyright Royal Society of Chemistry 2008. (**b**) High-throughput microfluidic single-cell RT-qPCR for gene expression analysis. Adapted by permission from Reference [110], copyright National Academy Sciences 2011. (**c**) Capturing single-cells along with a set of uniquely barcoded primers in microfluidic chip generated tiny droplets enables single-cell transcriptomics of many cells in a heterogeneous population. Adapted by permission from Reference [30], copyright Elsevier Inc. 2015.

**Figure 10 micromachines-10-00104-f010:**
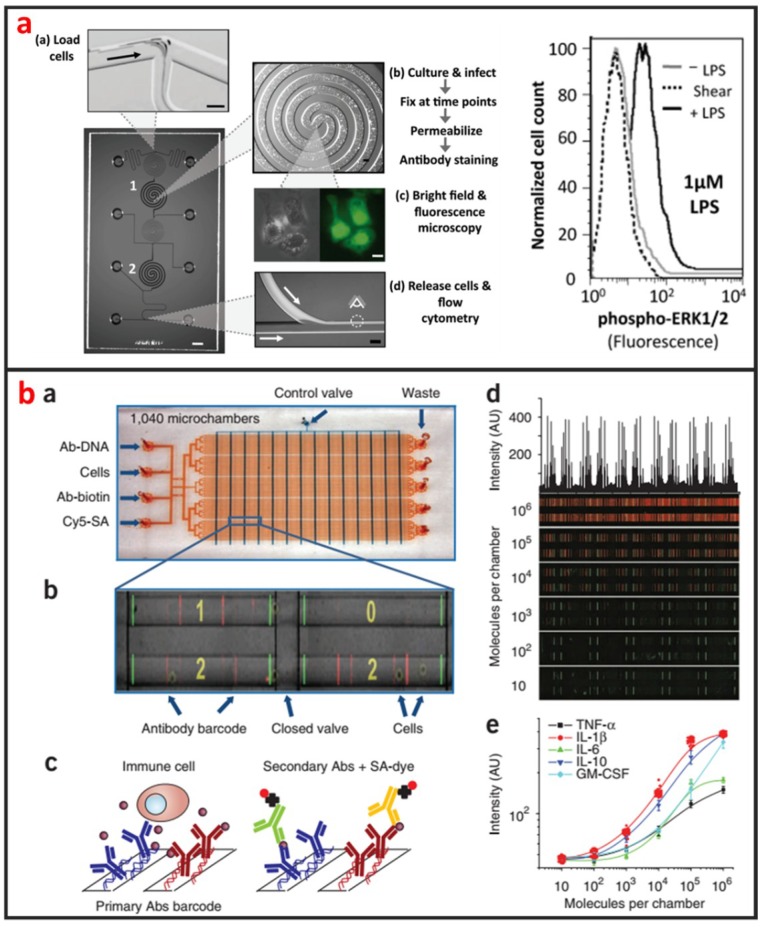
Single-cell protein analysis by using microfluidics. (**a**) A fully integrated microfluidic platform enabling automated phosphoprofiling of macrophage response. Adapted by permission from Reference [115], copyright American Chemical Society 2009. (**b**) A clinical microchip for detecting multiple cytokines of single immune cells reveals high functional heterogeneity in phenotypically similar T cells. Adapted by permission from Reference [116], copyright Nature Publishing Group 2011.

**Figure 11 micromachines-10-00104-f011:**
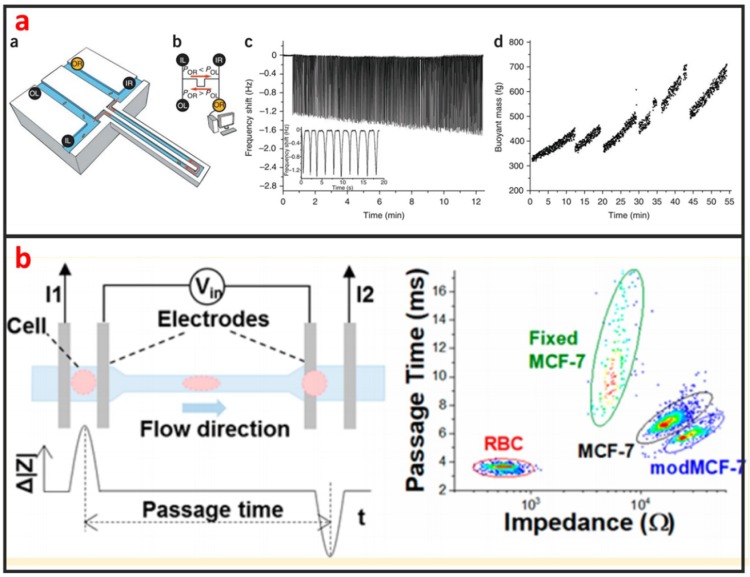
Biophysical property analysis of single-cells by using microfluidics. (**a**) A dynamic fluidic control system that enables the buoyant mass of cells as small as bacteria and as large as mammalian lymphocytes to be repeatedly measured with a suspended microchannel resonator (SMR). Adapted by permission from Reference [119], copyright Nature Publishing Group 2010. (**b**) A microfluidic device that can simultaneously characterize the mechanical and electrical properties of individual biological cells in a high-throughput manner (>1000 cells/min). Adapted by permission from Reference [127], copyright American Chemical Society 2018.

**Table 1 micromachines-10-00104-t001:** Various single-cell manipulations.

Manipulations	Descriptions
**Single-cell encapsulation** [26,27,28,29,30]	Entrapping single cells in isolated microenvironments
**Single-cell sorting** [12,31,32,33]	Separating homogenous populations of cells from heterogeneous populations at the single-cell resolution
**Single-cell trapping** [34,35,36]	Immobilizing single cells from bulk cells on the designated positions.
**Single-cell isolation** [37,38]	Pick or isolate single cells from bulk populations
**Single-cell rotation** [39]	Rotating targeted single cells
**Single-cell pairing** [20,40]	Positioning two homo- or heterotypic cells in proximity or contact
**Single-cell patterning** [23]	Positioning single cells on a substrate with defined spatial selection
**Single-cell stretching** [41]	Using external forces to deform targeted single cells
**Single-cell transportation** [42]	Moving cells at the single-cell level
**Single-cell lysis** [43,44]	Breaking down the targeted single cells
**Single-cell stimulation** [19,45,46]	Applying external physical/chemical/biological cues to stimulate targeted single cells

**Table 2 micromachines-10-00104-t002:** Comparison of various methods for single-cell manipulation.

Methods		Advantages	Disadvantages	Characteristics		
				Throughput	Efficiency	Accuracy
**Hydrody-namic method**	Droplet [27]	High-throughput, simple chip structure with great flexibility	Difficult to culture adherent cell, difficult to introduce biochemicals into droplets	250 µL/h	75%	--
	Inertial [55]	High throughput, high cell viability, and simple chip	Only work well under specific flow rates and cell concentrations	3 mL/min	84%	--
	Vortex [58]	Has no strict requirement about the properties of cells and fluid	Require external controller, low single-cell efficiency	--	--	Cell rotation 3.5 ± 2.1° s^−1^
	Micro-valve [60]	Reliable and fast for control, suitable for large-scale integration	Require complex and cumbersome external control devices	96 cells/chip	90.6 ± 8%	--
	Micro-structure [60]	Simple for operation, high throughput	Inflexibility, hard to control specific single-cell	10000 cells/chip	90%	--
**Electrical method**	Dielectro-phoresis [20]	Contactless, high selectivity, label-free	Require low-conductivity buffer	3264 pair of cells/chip	74.2%	--
	Electro-osmosis [70]	Label-free, easy for integrated fabrication	Low efficiency and accuracy with increased flow rate	81 cells/chip	73%	--
**Optical method**	Optical tweezer [12]	High accuracy and efficiency	Low throughput, high cost	--	97%	98%
	ODEP [76]	Flexible virtual electrodes, label-free, simple and low-cost	Require low-conductivity solution, opaque substrate	Scalable	--	--
	Opto-thermocapillary [77]	Flexible, can pattern single cells in hydrogel with high viability	High cost for cumbersome peripherical optical system	Low	High	High
**Acoustic method** [23]		Noninvasive, label-free, good penetrability	Need piezoelectric substrate for chip fabrication	High	--	High
**Magnetic method** [81]		Reliable and highly efficient	Not label-free	200 µL/min	> 85%	> 80%
**Micro-robot-assisted method** [84]		High accuracy, flexible and controllable	Low throughput	Low	High	--

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
