# Peer review of "Microfluidic Single-Cell Manipulation and Analysis: Methods and Applications"

_micromachines, 2019, doi:10.3390/mi10020104_

Round 1
Reviewer 1 Report
The manuscript entitled "Microfluidic Single-Cell Manipulation and Analysis: Methods and Applications" reviews various microfluidic methods and devices used for single cell analysis. The work is very interesting and informative. I would recommend the work to be published after a major revision, addressing my following comments:
1. Please provide a table summarising various cell manipulations, including sorting, trapping, stretching, etc. with a short description about each one.
2. Please provide a table, comparing various methods for manipulation of single cells, including hydrodynamic, dielectrophoresis, magnetophoresis, etc.
3. Please discuss and properly cite the following work, which describes a self-sufficient droplet generation system for studying the viability of cells:
A self-sufficient micro-droplet generation system using highly porous elastomeric sponges: A versatile tool for conducting cellular assays
Sensors and Actuators B: Chemical, 2018, 274, 645-653
4. In the conclusions, please provide a few recommendations in the form of bullet points for further advancement of this field.
Author Response
The revision parts of the review paper were highlighted in the revised version.
The detailed responses to the editors/ reviewers’ comments are attached.

Reviewer 2 Report
This review paper has three main parts, which are introducing different approaches of single-cell manipulation, discussing the applications of it and finally summarizing the advantages and disadvantages. The review is overall adequate and comprehensive.
Some additional comments to the authors:
1. The definitions of active and passive manipulation are not very clear. The definitions are given in line 95 and 97, which state "The active manipulation uses external forces generated by active control to ..." and "The passive manipulation uses interactive coupling between micro structure and fluid ...", respectively. However, the citations are not necessary to fit the definition. For example, inertial microfluidic approaches are apparently requiring flow control to achieve to certain Reynolds number for the sorting [32][35] and vortex generation also needs the external forces by active control of pillar movement [39]. Also, the difference between the works in 2.2.2 and 2.1.2 are not clear. This reviewer suggests the authors providing more concrete definitions for the categorization, or simply removing such a categorization.
2. When it comes to manipulation, speed and accuracy are two very critical properties. The authors are suggested to tabulate the speed, accuracy and other important properties of the cited works. It will be interesting for readers to know how fast and how precise have the cutting-edge techniques achieved.
For the purpose, this reviewer would like to suggest some recent publications on high-speed manipulations to the review paper:
(1) Sakuma et al., "On-chip cell sorting by high-speed local-flow control using dual membrane pumps", Lab Chip, 17, 2760-2767 (2017)
(2) Mizoue et al., "Transfer Function of Macro-Micro Manipulation on a PDMS Microfluidic Chip", Micromachines, 8(3), 80 (2017)
(3) Monzawa et al., "On-chip actuation transmitter for enhancing the dynamic response of cell manipulation using a macro-scale pump", Biomicrofluidics 9, 014114 (2015)
3. The authors are suggested to tabulate the main advantages and disadvantages of different approaches. They are now spreading in the contents and it is difficult for the readers to have a global vision of all of them. For example, hydrodynamic method has the advantage of high through-put (line 113). microvalve has the advantage of small size, fast response ... (line 180). optical tweezers have the advantage of high accuracy ... (line 348). Acoustics has advantages of good penetrability (line 403, 431 repeats again)
4. For echoing the statement in the abstract that "no two leaves are alike ... no two cells in a genetically identical group are the same ..." The authors are suggested to compare the single-cell analysis with conventional petri-dish experiments. What are the new insights that can be obtained with single-cell techniques?
Author Response

(The authors gave the same response as above.)

Round 2
Reviewer 1 Report
The authors have applied my previous comments. The manuscript can be published in its current form.
Reviewer 2 Report
The authors have properly addressed all my concerns and I recommend to accept the paper in its current form.